# DaMo: Data Mixing Optimizer in Fine-tuning Multimodal LLMs for Mobile Phone Agents

## Abstract

Mobile Phone Agents (MPAs) have emerged as a promising research direction due to their broad applicability across diverse scenarios. While Multimodal Large Language Models (MLLMs) serve as the foundation for MPAs, their effectiveness in handling multiple mobile phone tasks simultaneously remains limited. Although multitask supervised fine-tuning (SFT) is widely adopted for multitask learning, existing approaches struggle to determine optimal training data compositions for peak performance. To address this challenge, we propose DaMo (Data Mixture Optimizer) – a novel solution employing a trainable network that predicts optimal data mixtures by forecasting downstream task performance for any given dataset ratio. To support comprehensive evaluation, we introduce PhoneAgentBench, the first specialized benchmark to evaluate MLLMs on multimodal mobile phone tasks, comprising 1,235 QA pairs spanning diverse real-world industrial mobile application scenarios. Demonstrating strong predictive capability ($R^2$=0.81) in small-scale pilot experiments, DaMo efficiently extrapolates optimal data mixing configurations. Our results show DaMo achieves a 3.38% performance improvement on PhoneAgentBench compared to alternative methods. Furthermore, extensive experiments across established benchmarks including BFCL-v3, MME-Reasoning, MME-Perception, and OCRBench reveal DaMo's superior generalization, outperforming other approaches by 2.57% in terms of average score. When used solely for MLLM optimization on the BFCL-v3 task, DaMo improves the metrics by 12.47% than other methods. Notably, DaMo maintains robust scalability, preserving its effectiveness when applied to other model architectures.

## 1 Introduction

Mobile phone agents (MPAs) have attracted huge attention due to their practicability in a multitude of scenarios. An ideal MPA has to master multiple capabilities, such as environment perception Zhang et al. (2024); Ingold (2021), task planning Song et al. (2023); Liu et al. (2024b), multimodal reasoning Lu et al. (2022); Wang et al. (2024), function call Chen et al. (2024a); Basu (2024), and personalized memory Li et al. (2024a); Yuan et al. (2023).

The advent of multimodal large language models (MLLMs) provides a promising solution for the ideal agent. However, existing MLLMs encounter significant challenges in effectively integrating these diverse capabilities. Consequently, developing a versatile model capable of handling multiple tasks is critical for creating a advanced phone agent.

Multitask supervised fine-tuning (SFT) is the predominant approach utilized to empower MLLMs in addressing multiple tasks. Nevertheless, in light of numerous training datasets and downstream tasks, identifying optimal data blending strategies to maximize model performance remains a significant research challenge. The existing works on data mixture optimization Xie et al. (2023b); Ge et al. (2024); Albalak et al. (2023) focus on the pretraining phase by predicting validation loss. However, these methods are inadequate to determine the optimal data mixture for multitask SFT, as they fail to directly correlate with model performance on downstream tasks.

We investigate whether downstream task performance can be reliably predicted for any given data mixture prior to actual model training, including identifying the optimal mixture that would yield optimal performance. To this end, we propose the **d**ownstream task **p**erformance **p**rediction (DaPP) method to build **Da**ta **M**ixing **O**ptimizer (DaMo). DaPP leverages a function to straightly predict

Figure 1: Illustration of our pipeline for obtaining the optimal data mixture. Left: Given $m$ training sets with a batch size of $b$, all possible mixture combinations constitute the data mixing space. We sample a small number of data mixture from this space, train them on a small MLLM, and then evaluate downstream task performance. Using the data mixture as inputs and the metrics as outputs, we fit a MLP to establish the DaMo. By extrapolating from the data mixing space, we predict the optimal data mixture to train the MLLM. Right: Demonstrates the extension and alignment of DaMo to other MLLMs and new data mixing spaces.

model performance at downstream tasks. Considering that exponential functions used in Xie et al. (2023b); Ge et al. (2024); Albalak et al. (2023) are not well-aligned with SFT performance trajectories for specific downstream applications Huang et al. (2019); Xie et al. (2024); Isik et al. (2024), we propose to utilize a trainable neural network for target fitting. The optimal data mixture is obtained through extrapolation via DaMo, with the process shown in Fig. 1.

Another obstacle in developing an ideal mobile phone agent is the absence of comprehensive real-world industrial benchmarks for evaluating MPA performance. Current benchmarks Gao et al. (2024); Cheng et al. (2024); Li et al. (2025); Wang et al. (2025) in this domain predominantly focus on Graphical User Interface (GUI) tasks, which fail to capture the full spectrum of practical application scenarios. To address this critical gap, we introduce PhoneAgentBench - a thorough benchmark encompassing four fundamental capabilities: 1) complex task planning, 2) device-native tool usage, 3) multimodal memory, and 4) screen context understanding. Our benchmark comprises 2,350 meticulously validated test cases that simulate real-world phone interactions, with 55% necessitating the simultaneous activation of more than 3 tools.

Our proposed DaMo demonstrates three key advantages. First, it achieves 3.38% average performance gain on PhoneAgentBench compared to state-of-the-art method, DML Ye et al. (2024). Second, when evaluated on the general benchmarks including BFCL-v3 Yan et al. (2024), MME-perception Fu et al. (2023), MME-reasoning Yuan et al. (2025), and OCRBench Liu et al. (2024c), DaMo outperforms DML by 2.57% in terms of average score. Third, DaMo exhibits robust scalability—Pearson correlations between predicted and actual scores remaining consistently high (0.75~0.95) across other models, while introducing significant gains on downstream tasks over other methods.

Our core contributions are as follows.

- We propose Downstream Task Performance Prediction method to establish a Data Mixing Optimizer, which directly estimates model performance on downstream tasks for optimal data mixing.

- We construct PhoneAgentBench, a benchmark spanning four critical dimensions: complex task planning, device-native tool usage, multimodal memory, and screen context understanding, mirroring real-world mobile interaction scenarios.

- Through systematic experiments, our method demonstrates exceptional generalization and scalability, outperforming other methods on PhoneAgentBench, and achieving state-of-the-art performance on the BFCL-V3 leaderboard among 4B-scale models, while also maintaining stable prediction accuracy with efficient adaptation to other models.

Table 1: PhoneAgentBench

| Dataset | Evaluation ability | Data size | Dataset | Evaluation ability | Data size |
|---------|-------------------|-----------|---------|-------------------|-----------|
| MT-Plan | Mulitmodal Task Planning | 100 | MM-RR | Multimodal Reference Resolution | 130 |
| ACU | Agent Context Understand | 100 | MM-NER | Multimodal Named Entity Recognition | 376 |
| APP-Rec | APP Recognition | 100 | Mobile-FC | Mobile Function Calling | 429 |

## 2 RELATED WORK

**Data Mixing** Data mixtures follow distinct optimization paths in pre-training versus fine-tuning. Current research can be broadly divided into two paradigms. For pre-training, methods optimize language model perplexity (PPL). Early heuristic approaches like uniform sampling Michel et al. (2021) gave way to learnable solutions; DoReMi Xie et al. (2023a) uses Group DRO Sagawa et al. (2020) for domain weights; ODM Albalak et al. (2023) frames selection as a bandit problem; BiMix Ge et al. (2024) jointly optimizes domain proportions and data scaling. All these PPL-focused approaches differ fundamentally from fine-tuning objectives.

Fine-tuning research remains limited: industrial solutions (LLaMA3 Grattafiori et al. (2024), Tulu3 Lambert et al. (2024)) rely on costly manual iteration; SFTMix Xiao et al. (2024) optimizes intra-dataset ratios but cannot handle multi-source data; MoE-based methods Zhu et al. (2024) adjust weights but lack interpretable criteria. Key gaps remain in developing general multi-source mixing schemes and theoretical guidance for fine-tuning datasets.

**Agent Benchmark** Recent advancements in agent-based systems have spurred various benchmarks to evaluate their capabilities. PlanBench Valmeekam et al. (2023) and REALM-Bench Geng & Chang (2025) assess planning capabilities. ToolBench Qin et al. (2023), BFCL Yan et al. (2024), and API-Bank Li et al. (2023) evaluate tool invocation and ReflectionBench Li et al. (2024b) measures self-reflection. LTM Benchmark Castillo-Bolado et al. (2024) tests memory retention. These benchmarks are limited to single-dimensional evaluations, lacking holistic assessment. GAIA Mialon et al. (2023) uses end-to-end evaluation to assess general agents, but lacks granularity. AgentBench Liu et al. (2023) and KAgentBench Pan et al. (2023) are unimodal, ignoring multimodal interaction. Moreover, all of these benchmarks deviates from real-world phone scenarios. ScreenSpot-Pro Cheng et al. (2024), MobileViews Gao et al. (2024), VisualAgentBench Liu et al. (2024a), ScreenSpot-Pro Li et al. (2025), and MMBench-GUI L2 Wang et al. (2025) can evaluate phone agents, but they are designed mainly for GUI tasks in mobile scenarios. A critical gap remains: the absence of a comprehensive benchmark supporting multimodal interaction while systematically evaluating mobile phone agents across planning, tool usage, memory, and other dimensions. This hinders iterative optimization and underscores our work's innovation potential.

## 3 PHONEAGENTBENCH

Current open-source agent evaluation benchmarks Valmeekam et al. (2023); Li et al. (2024b); Liu et al. (2023); Geng & Chang (2025) are unable to assess agents for tackling multimodal tasks in mobile phone scenarios. Mobile Phone agent relevant benchmarks Gao et al. (2024); Cheng et al. (2024); Li et al. (2025); Wang et al. (2025) primarily evaluate the GUI tasks of MLLMs. However, they lack support for multimodal interaction and do not provide systematic evaluation across key dimensions such as planning, tool use, and memory. This mismatch with mobile phone agent scenarios poses a significant challenge. To address this gap, we aim to develop a benchmark tailored to real-world industrial application scenarios, thereby accelerating the practical implementation of agent technology.

We develop a novel benchmark suite specifically designed for mobile phone agents. This suite encompasses six carefully curated datasets focusing on key mobile phone application tasks, thereby offering a holistic assessment of phone agents' performance across diverse capabilities critical to real-world mobile applications. Details of the tasks are provided in Table 1. We describe the data construction process using the Multimodal Task Planning task (MT-Plan) as a case in point.

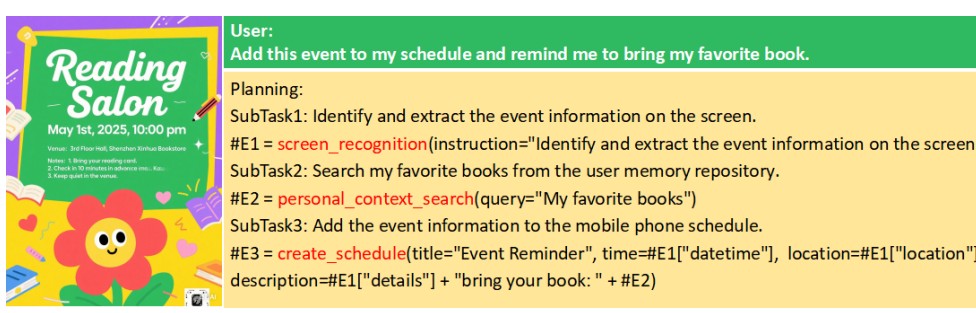

Figure 2: MT-Planning example

**MT-Plan**  MT-Plan is designed to evaluate multimodal task planning capabilities. Unlike T-Eval planning Chen et al. (2023), it focuses on multimodal complex task interactions in phone agent scenarios. As shown in Fig.2, MT-Plan takes <image + query> as input and outputs a planning structured as a directed acyclic graph (DAG). Images are sourced from real photos or mobile screenshots, while tools are derived from APIs provided by operating systems or app ecosystems. Queries and plannings are carefully constructed by annotators based on the images. Queries are required to be concise, colloquial, and aligned with real users' daily needs. Meanwhile, tasks must be sufficiently complex to require plannings to invoke at least 2 tools. To ensure data accuracy, three annotators were invited to conduct cross-validation. Additionally, to evaluate the dataset's complexity and diversity, we compared the metrics of MT-Plan and T-Eval planning, as presented in Table 7.

The MT-Plan evaluator adopts the T-Eval planning evaluator Chen et al. (2023): it compares the predicted plan with the golden plan, and calculates the score based on the length of the longest ordered action sequence derived from similarity-matched pairs.

The construction methods of the remaining five datasets are presented in AppendixA.2.

## 4  METHODOLOGY

This section formalizes multitask fine-tuning optimization as identifying the optimal data mixture to maximize downstream task metrics. We propose predicting unseen mixture performance by fitting the performance of downstream tasks with limited training configurations. Analysis of single and dual dataset experiments demonstrates why exponential/power-law functions fail to model convergence patterns, prompting our neural network solution for extrapolating optimal data mixture.

### 4.1  PROBLEM FORMULATION

Consider fine-tuning a MLLM using a mixture of $m$ heterogeneous training datasets, denoted as $\mathcal{D} = \cup_{i=1}^{m} \mathcal{D}_i$. Each $\mathcal{D}_i$ contains $n_i$ labeled samples with the total number of samples being $N = \sum_{i=1}^{m} n_i$. We fine-tune the MLLM starting from initial parameters $\theta_0$, using a batch size $b$, for a maximum of $T = \lceil N/b \rceil$ training steps.

We define the **data mixture proportion** as $\mathbf{p} = [p_1, p_2, ..., p_m]$, where $p_i$ represents the proportion of samples drawn from dataset $\mathcal{D}_i$. The data mixture proportion $\mathbf{p}$ satisfies $\sum_{i=1}^{m} p_i = 1$.

Similarly, we consider $k$ downstream test datasets, denoted as $\mathcal{D}^{test} = \cup_{j=1}^{k} \mathcal{D}_j^{test}$. Let $\mathbf{s} = [s_1, ..., s_k] \in [0,1]^k$ represent the score of each test dataset. The overall average score of the MLLM with parameters $\theta$ is given by $S_\theta = \frac{1}{k} \sum_{j=1}^{k} s_j$.

We aim to find the optimal data mixture proportion $\mathbf{p}^* \in \mathcal{P}$ (where $\mathcal{P}$ denotes the complete data mixing space, $\mathbf{p} \in \mathbb{R}^m$) that maximizes the overall average score of downstream tasks:

$$\mathbf{p}^* = \arg\max_{\mathbf{p} \in \mathcal{P}, t \leq T} \mathbb{E}_{\theta \sim \mathcal{A}(\mathbf{p}, t, \theta_0)} S_\theta \tag{1}$$

where $\mathcal{A}$ denotes the fine-tuning process that produces the MLLM's parameters $\theta$ based on the initial parameters $\theta_0$ for $t$ steps using the data mixture strategy $\mathbf{p}$.

Without any constraints, the size of the set $\mathcal{P}$ that represents batch-wise permutations is given by $|\mathcal{P}| = \frac{N!}{(b!)^T}$, which is computationally intractable. Therefore, we introduce some necessary assumptions to prune the space $\mathcal{P}$. By disregarding the order of samples within the same dataset and keeping the data mixture fixed throughout the entire number of training steps $T$, we obtain a smaller data mixing space $\mathcal{P}_{fix}$. According to the principle of combination with repetition, the size of this fixed data mixing space $\mathcal{P}_{fix}$ is given by $|\mathcal{P}_{fix}| = C^m_{m+b-1}$.

## 4.2 PERFORMANCE PREDICTION OF DOWNSTREAM TASKS

We aim to find the optimal mixture $\mathbf{p}^* \in \mathcal{P}$. Given the high training cost of MLLMs, an exhaustive brute-force search is clearly impractical. To address this problem, we propose DaMo which is able to estimate model performance at downstream tasks without training, given any mixture proportions of training data. Towards this target, we fit a function $f$ to predict performance based on data mixtures. To obtain accurate $f$, a efficient sampling approach is proposed to generate training samples. The sampling process is detailed as: 1) Randomly select a small set of m-dimensional mixing ratios from $\mathcal{P}_{fix}$. 2) Train MLLM while saving checkpoints at every $\tau$ steps. 3) Evaluate each checkpoint to obtain the performance of downstream tasks. This process yields the mapping: (data mixture, training steps) $\rightarrow$ performance of downstream tasks. Based on these samples, we fit $f$ to predict the performance trajectory of unseen mixture:

$$\hat{\mathbf{s}} = f(\mathbf{p}, t; \theta_0), \qquad (2)$$

where $\theta_0$ is initial model state and $t = \tau * i$ is train steps of the i-th checkpoint. With an accurate fitting of $f$, we can extrapolate performance estimates across the entire $\mathcal{P}_{fix}$ space, dramatically reducing the model training costs required to identify the optimal data mixture.

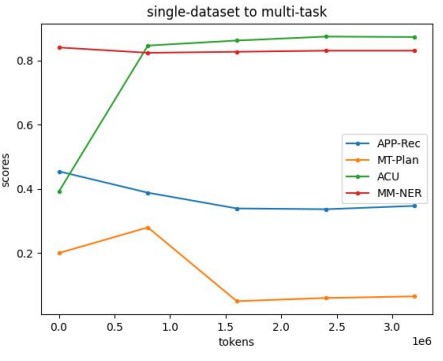

(a) Performance under single-dataset training

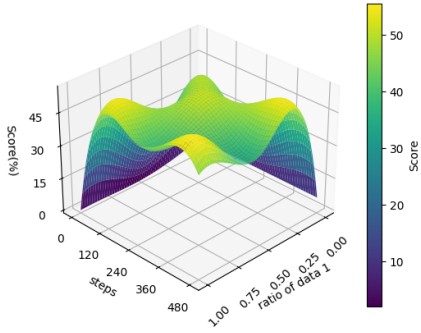

(b) Performance under dual-dataset mixtures

Figure 3: Training dynamics on downstream tasks

The critical challenge lies in selecting an appropriate function $f$. While conventional exponential or power-law functions Achiam et al. (2023); Grattafiori et al. (2024) are widely adopted for pretraining loss convergence, we hypothesize their inadequacy in multi-task fine-tuning scenarios involving interacting datasets. To validate this, we systematically analyze training dynamics under two configurations: (1) single-dataset training (MultiModal-Understanding, MMU) and (2) dual-dataset mixtures (APP Recognition (APP-Rec) + MMU, see Section 5.1).

We trained a MLLM on the MMU dataset and evaluated its performance on PhoneAgentBench. As shown in Fig. 3(a), the results reveal distinct task-specific patterns: (1) **Enhancement**: MMU significantly improves ACU performance. (2) **Conflict**: APP-Rec performance degrades with MMU training steps. (3) **Neutrality**: MM-NER shows no correlation with MMU training. (4) **Overfitting**:

MT-Plan exhibits initial gains followed by sharp declines, indicating harmful overfitting beyond optimal data volume.

Fig. 3(b) demonstrates the complex interaction when training on the mixed dataset of APP-Rec and MMU for the APP-Rec task. The 3D performance surface (X: training steps, Y: APP-Rec training dataset ratio, Z: APP-Rec bench score) exhibits **non-convex topology with non-monotonic fluctuations** along both axes. This nonlinearity fundamentally prevents analytical solutions for Eq. 1 and invalidates conventional exponential and power functions.

Motivated by neural networks' capacity to model high-dimensional nonlinearities, we pioneer their application to DaMo. Our framework implements $f$ as a multi-layer perceptron (MLP) that directly maps data mixture and training step to task performance:

$$\hat{\mathbf{s}} = f_{MLP}(\mathbf{p}, t; \theta_0),\tag{3}$$

### 4.3 Optimal Data Mixture Extrapolation

When we define the data mixture space as $\mathcal{P}_{fix}$ and employ MLP as the fitting function, the optimization objective in Eq. 1 can be reformulated as follow.

$$\mathbf{p}_{\text{fix}}^* = \arg\max_{\mathbf{p}\in\mathcal{P}_{\text{fix}},t\leq T} \frac{1}{k}\sum_{j=1}^{k} f_{MLP}^j(\mathbf{p}, t; \theta_0).\tag{4}$$

Where $j$ denotes $j$th downsteam task. Given the negligible inference cost of MLP models, DaMo can efficiently extrapolate the optimal data mixture. We first iterate through all possible data mixtures in the $\mathcal{P}_{fix}$ space to predict downstream task performance scores. Subsequently, we sort these predicted scores and select the top-k highest-scoring mixtures to train our MLLM. This approach enables us to systematically identify the optimal data mixture without exhaustive empirical testing. The complete algorithm pseudocode is provided in Appendix B.

## 5 Experiments

### 5.1 Experiments Settings

**Training datasets**  Our training data corpus integrates 12 open-source and self-built datasets, encompassing both Chinese and English languages. The comprehensive dataset contains a total of 220K instructions, providing a diverse and extensive resource for our model training. Details are provided in Appendix A.3

**Downstream Task Evaluation**  Besides PhoneAgentBench, we further evaluated our method on four widely used open-source benchmarks to verify generalization, including BFCL-V3 Yan et al. (2024), MME-perception Fu et al. (2023), MME-reasoning Yuan et al. (2025) and OCRBench Liu et al. (2024c). These benchmarks collectively encompass a total of ten evaluation tasks, and all metrics are expressed as percentages (0-100%), with higher values indicating superior performance.

**Baseline**  We selected two heuristic approaches commonly used in industry and one representative loss-based exponential fitting method: **Uniform Mixture:** All datasets are sampled with equal weights. **Natural Mixture:** Sampling weights are proportional to the size of each dataset. **Data Mixing Laws (DML)** Ye et al. (2024): An exponential function-based loss fitting method to predict the optimal mixture.

**Implement Details**  We conducted a series of experiments to verify the effectiveness of DaMo. Initially, we performed training and evaluation based on InternVL2.5-4B Chen et al. (2024b) to obtain fitting samples (in the format of $(\mathbf{p}, t, \mathbf{s})$) for a two-layer MLP training. Subsequently, we calculated the coefficient of determination ($R^2$) Wright (1921) via 10-fold cross-validation to verify the fitting performance. Then, we used the MLP to predict the downstream task performance of unseen data mixtures and train MLLM on the optimal data mixture. Finally, we verified the scalability of DaMo on other models. Additional implement details are available in Appendix A.1.

## 5.2 FITTING SCORE OF NEURAL NETWORK

As analyzed theoretically for $\mathcal{P}_{\text{fix}}$ in Section 4.1, when the number of training datasets $m = 12$ and batch size $b = 16$, $\mathcal{P}_{\text{fix}}$ forms a discrete enumerable space with a size of $\binom{12+16-1}{12} \approx 1.7 \cdot 10^7$. This is a fairly large space, so how many sample points does DaMo need to fit the entire $\mathcal{P}_{\text{fix}}$ well? We gradually increased the number of training sample points for the MLP, and as shown in Table 2, when the number reaches 250, the fitting score of the MLP gradually converged. Considering the training cost, we stopped further experiments. Notably, 250 samples account for only a negligible portion of the entire $\mathcal{P}_{\text{fix}}$ space, yet they enable $R^2 = 0.81$ in 10-fold cross-validation. This indicates that the performance of MLLM on downstream tasks has an inherent connection with the characteristics and mixing patterns of training data, and DaMo learns this mapping via neural networks.

Table 2: MLP fitting dynamics

| Number of fitting samples | Cost of getting samples (H20-hours) | Score ($R^2$) |
|---|---|---|
| 50 | 872 | 0.58 |
| 100 | 1817 | 0.57 |
| 150 | 2581 | 0.74 |
| 200 | 3521 | 0.78 |
| 250 | 4225 | 0.81 |

## 5.3 DOWNSTREAM TASK PERFORMANCE OF UNSEEN DATA MIXTURES

We commence with a systematic analysis of the sample point distribution. As shown in Fig. 4(a), the score distribution under random data mixtures approximately follows a normal distribution, revealing two critical characteristics: (1) The absence of a right-side long tail indicates that excellent data mixtures are extremely sparse. (2) The performance of random mixture is predominantly mediocre, and baseline methods (vertical dashed line) show no discernible advantage, demonstrating the inefficiency of heuristic approaches.

We used DaMo to predict across $\mathcal{P}_{\text{fix}}$ space, selected the top 50 data mixtures with the best predicted performance, and conducted actual training and evaluation on MLLM. The distribution of their performance is shown in Fig. 4(b), DaMo successfully identifies data mixtures with significantly higher overall average scores compared to baseline methods.

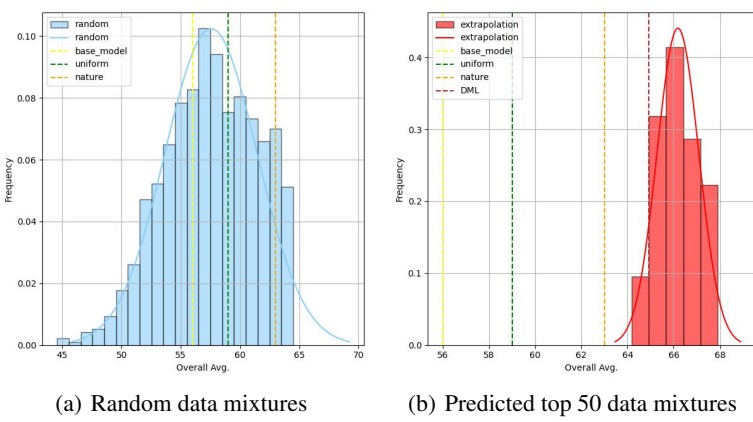

(a) Random data mixtures      (b) Predicted top 50 data mixtures

Figure 4: Probability distributions of overall average scores across different checkpoints.

Through selecting mixture with top 1 predicting score on PhoneBenchAgent and open-source benchmarks to train MLLM, we obtain the performance on PhoneBenchAgent and open-souce benchmarks, as shown in Table 3. DaMo achieves more than 23% (from 44.83% to 68.18%) improvement over the native model (without SFT) on PhoneAgentBench, surpassing both uniform and natural mixture strategies. When general capabilities are concurrently considered, DaMo enhances domain-specific

Table 3: Main results on PhoneAgentBench and open-source benchmarks by using top-1 data mixture to train MLLM, predicted by DaMo on PhoneAgentBench and open-source benchmarks.

| Method | MT-Plan | APP-Rec | MM-RR | ACU | MM-NER | Mobile-FC | OS Avg. | PAB Avg. | Overall Avg. |
|---|---|---|---|---|---|---|---|---|---|
| w/o SFT | 20.00 | 6.00 | 65.38 | 39.18 | **84.08** | **54.31** | **68.77** | 44.83 | 54.40 |
| uniform | 54.50 | **56.00** | 44.62 | **86.37** | 81.71 | 45.92 | 55.76 | 61.52 | 59.22 |
| natural | 47.00 | 46.00 | 86.15 | 83.10 | 79.83 | 49.88 | 59.95 | 65.33 | 63.18 |
| DML Ye et al. (2024) | 52.00 | 43.00 | 85.38 | 85.72 | 80.01 | 42.66 | 65.48 | 64.80 | 65.07 |
| DaMo | **55.50** | 51.00 | **86.15** | 85.30 | 83.34 | 47.79 | 68.05 | **68.18** | **68.13** |

Table 4: Main results on open-source benchmarks of MLLMs trained by predicted optimal data mixture on open-source benchmarks.

| Method | BFCL-V3 | MME-perception | MME-reasoning | OCRBench | OS Avg. |
|---|---|---|---|---|---|
| w/o SFT | 29.32 | 83.82 | 79.42 | 82.50 | 68.77 |
| uniform | 34.69 | 58.63 | 64.91 | 64.80 | 55.76 |
| natural | 31.41 | 75.47 | 67.01 | 65.90 | 59.95 |
| DML Ye et al. (2024) | 25.47 | 83.31 | 76.34 | 76.8 | 65.48 |
| DaMo | 43.15 | 84.53 | 80.94 | 83.60 | **73.06** |
| DaMo (∗) | **47.43** | **85.12** | **82.54** | **83.90** | / |

∗: These scores correspond to different checkpoints, which are optimized by DaMo on a single task.

capabilities on the PAB while preserving generalizability, yielding an overall average score improvement of 13.73%. Compared to DML Ye et al. (2024), we observe stable performance gains across almost all tasks, which validates the advantage of fitting the relationship between data mixtures and downstream performance.

To study the generalization of DaMo on general tasks, we employ DaMo to predict MLLM's performance only on open-source benchmarks, and use the top-1 data mixture to train MLLM, reporting the results in Table 4. It can be observed that our DaMo achieves remarkably superior performance across all open-source benchmarks compared to baselines. It is noteworthy that focusing on task-specific objectives leads to significantly greater improvements. This is clearly demonstrated by the performance growth from 29.32% to 47.43% on the BFCL-V3 benchmark, implemented by DaMo (BFCL-V3) which predicts the performance on BFCL-V3 benchmark only to search optimal data mixture. Crucially, this enhancement is sustained even in the absence of any task-curated training data. We posit that the observed performance benefit is fundamentally driven by DaMo's methodology of exploring optimal mixtures, which orchestrates a balanced advancement across both specialized and generalizable capabilities.

## 5.4 EXTENSION TO OTHER MODELS

We are concerned with the effective generalization of the DaMo to other models. Most current work on data mixture during the pretraining phase assumes that data mixture strategies can be directly transferred from smaller models to larger ones Xie et al. (2023b), but their applicability in the supervised fine-tuning phase remains unverified. To this end, we conducted experiments on transferring DaMo obtained from InternVL2.5-4B to Qwen2.5VL-3B-Instruct, Qwen2.5VL-7B-Instruct Bai et al. (2025), and InternVL3-14B Zhu et al. (2025) with zero or minimal additional training cost.

Table 5: Main results of scalability testing on PhoneAgentBench and open-source Benchmarks

| Model | w/o SFT | uniform | natural | DML | DaMo (orig.) | DaMo (lin.) |
|---|---|---|---|---|---|---|
| Qwen2.5VL-3B-Inst. | 56.25 | 65.15 | 64.82 | 65.03 | 68.02 | 68.66 |
| Qwen2.5VL-7B-Inst. | 59.43 | 68.48 | 65.99 | 66.37 | 67.79 | 69.09 |
| InternVL3-14B | 67.84 | 63.56 | 67.8 | 66.45 | 68.86 | 69.75 |

We randomly selected a subset of data mixtures from the training and extrapolation samples of the original DaMo and used them to train the new model. As shown in the upper panel of Fig. 5, the Pearson correlation coefficients ($r$) are generally above 0.75, demonstrating the robust cross-model applicability of DaMo. This suggests that optimal mixtures identified for the base model likely remain near-optimal for the target models.

Due to the varying capabilities across different models, directly transferring DaMo to other models introduces prediction biases. Therefore, we regard DaMo as a model-agnostic predictor: for a new model, we train 20 calibration samples to fit a linear layer that compensates for model discrepancies. The linear-mapped DaMo is defined as $g = f(.)\mathbf{w} + b$ (see details in Appendix C). As shown in the bottom panel of Fig. 5, after applying this linear mapping, the discrepancies between models are reduced, leading to a further enhancement in correlation with $r$ increasing to above 0.9.

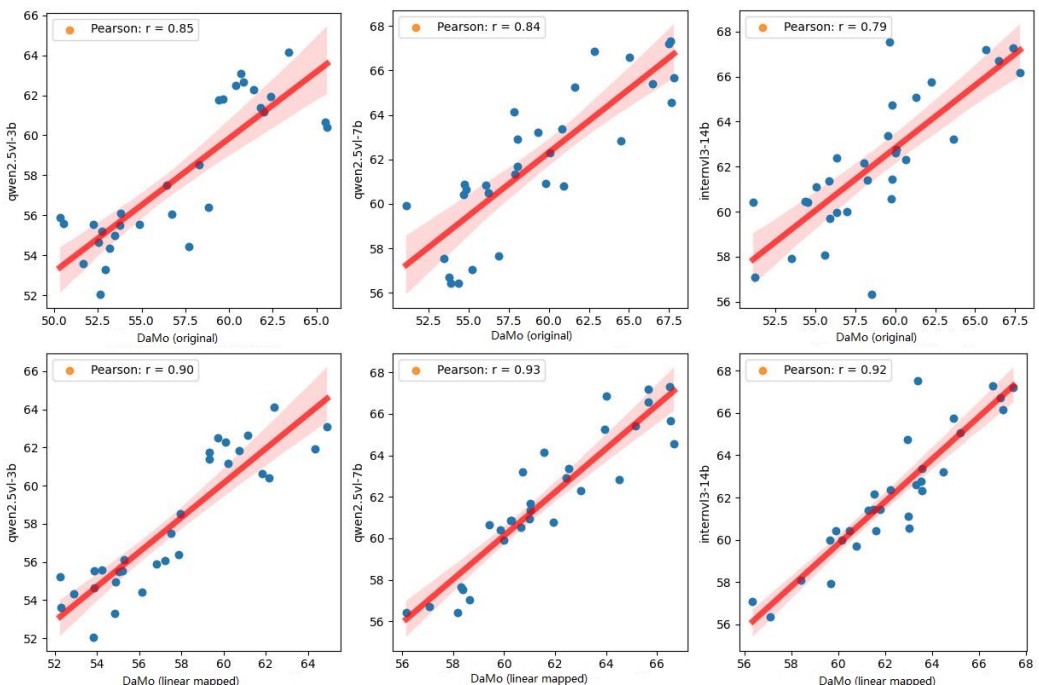

Figure 5: Scalability Analysis. Top: Scatter plot comparing the predicted overall average scores by the original DaMo against the actual scores of target models. Bottom: Apply linear-mapped correction to DaMo.

We performed training and evaluation on the new models using the optimal data mixture predicted by DaMo. As shown in Table 5, directly applying the original DaMo achieves competitive overall average scores, demonstrating the stable transferability of DaMo. Using the linear-mapped DaMo, the scores can be further improved, indicating that the linear mapping mitigates differences in model capabilities and better aligns DaMo with target models.

## 6 CONCLUSION

In this paper, we present the Data Mixing Optimizer (DaMo) to optimize data mixtures in multitask fine-tuning of multimodal large language models. By introducing downstream task performance prediction with neural network-based modeling, DaMo can predict model performance for any given data mixture. To support comprehensive evaluation, we introduce PhoneAgentBench for evaluation of multimodal large language models on phone agentic tasks. Moreover, DaMo can be extended to other models and tasks. Experimental results demonstrate the efficacy of DaMo not only on PhoneAgentBench, but also on general benchmarks, outperforming the state-of-the-art methods.

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

# A   DETAILS OF EXPERIMENTS SETTING

## A.1   IMPLEMENT DETAILS

We applied a series of experiments to verify the effectiveness of DaMo. Initially, we conducted training and evaluation on InternVL2.5-4B Chen et al. (2024b) to obtain fitting samples for the MLP. Specifically, we first sampled 250 random data mixtures $\mathbf{p}$ from $\mathcal{P}_{\text{fix}}$. For each mixture, training was performed on 8 NVIDIA H20 GPUs, and checkpoints were saved at 4 distinct training steps—resulting in a total of 1000 checkpoints. All 1000 checkpoints were then evaluated on downstream tasks, which generated 1000 sample points in the format of $(\mathbf{p}, t, \mathbf{s})$. The hyperparameters for training the MLLM are listed in Table 6.

Subsequently, we fitted the MLP on these 1000 sample points. MLP is structured as a two-layer multi-layer perceptron (MLP) built upon `sklearn.MLPRegressor`, where each of the two hidden layers contains 100 neurons. To verify the model's fitting score, we assessed the coefficient of determination ($R^2$) Wright (1921) of DaMo via 10-fold cross-validation. More details of MLP are provided in Table 6.

Then, we utilized DaMo to predict the downstream task performance of unseen data mixtures. Leveraging the low inference cost of the MLP, we conducted performance predictions for all mixtures $\mathbf{p} \in \mathcal{P}_{\text{fix}}$. Among these, the 50 data mixtures with the optimal predicted performance were selected for further model training and validation, aiming to obtain actual performance metrics.

Finally, to verify the scalability of DaMo on other models, we extended DaMo (based on InternVL2.5-4B) to Qwen2.5VL-3B-Instruct, Qwen2.5VL-7B-Instruct Bai et al. (2025), and InternVL3-14B Zhu et al. (2025). For these new models, we trained a small number of random mixtures, analyzed the correlation between DaMo's predicted performance and the actual training performance, and meanwhile used DaMo to find the optimal mixtures on the new models to verify whether it still maintains competitiveness compared with the baselines.

Table 6: Hyperparameters of training

| Model | Hyperparameters | setting |
|---|---|---|
| MLLMs | AdamW $\beta_1$ | 0.9 |
| | AdamW $\beta_2$ | 0.95 |
| | AdamW $\epsilon$ | $1e-6$ |
| | Max Sequence Length | 16384 |
| | Batch Size | 16 |
| | Gradient Accumulation Steps | 8 |
| | Training Steps | 1440 |
| | Warmup Steps | 144 |
| | Peak Learning Rate | $1e-5$ |
| | Weight Decay | 0.1 |
| | Gradient Clipping | 1.0 |
| MLP | Input Layer Dimension | 12 |
| | Hidden Layer 1 Dimension | 100 |
| | Hidden Layer 2 Dimension | 100 |
| | Output Layer Dimension | 10 |
| | Activation Function | ReLU |
| | Optimizer | Adam |
| | Learning Rate | $1e-6$ |
| | Training Steps | 10000 |

## A.2   EVALUATION DATASETS

To guarantee the faithfulness of the proposed PhoneAgentBench, we implemented a rigorous workflow encompassing data filtering, synthetic data generation, and manual verification. Details information about our evaluation datasets for PhoneAgentBench are as follows.

### A.2.1 MULTIMODAL TASK PLANNING

We introduce the two metrics of complexity and diversity to evaluate the quality of the benchmark for the task planning.

- **Complexity**: The answer of MT-Planning can be viewed as a directed acyclic graph (DAG), where each subtask is a node and the dependency relationship between subtasks are edges. Thus, complexity can be expressed as $n_{edge}/n_{node}$.

- **Diversity**: The higher the similarity between queries in a dataset, the lower the diversity of that dataset. We use Rough-L to calculate the similarity between every pair of queries, and diversity can be expressed as $1 - \frac{1}{N(N-1)/2} \sum_{i \neq j} \text{Rough-L}(q_i, q_j)$.

Based on this, we compared the data complexity and diversity between MT-Plan and T-Eval planning.

Table 7: benchmark metrics

| Benchmark | complexity↑ | diversity ↑ |
|---|---|---|
| MT-Plan | 0.661 | 0.82 |
| T-Eval planning Chen et al. (2023) | 0.122 | 0.73 |

### A.2.2 MULTIMODAL REFERENCE RESOLUTION

The MultiModal Reference Resolution (MM-RR) task requires the model to determine whether the current question refers to information in the image, which is a binary classification task. As shown in Fig. 6, the question in Fig. 6(a) does not refer to the content in the image, so the answer is 0; while the question in Fig. 6(b) refers to the drinks on the shelf in the image, so the answer is 1.

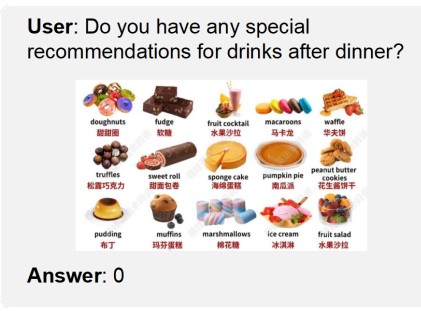
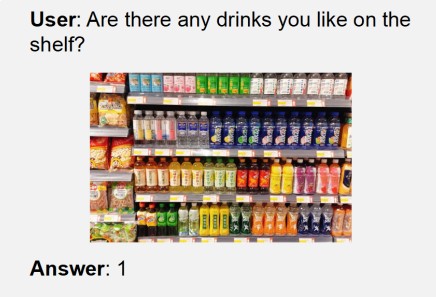

(a) A negative sample example.    (b) A positive sample example.

Figure 6: Examples of RR dataset.

### A.2.3 MULTIMODAL NER

Multimodal NER (MM-NER) benchmark quantitatively measures MLLMs' ability in understanding and extracting key entities. The dataset comprises 376 image-only samples sourced from Baidu's publicly available image repositories, where each image underwent a rigorous curation process: professional annotators manually filtered the raw visual data to retain high-quality, clearly discernible images, which were subsequently annotated with precise labels focusing on seven critical entity categories—temporal references, geographical locations, personal identifiers, contact numbers, tracking Number, flight Number, train Number to establish a structured benchmark for multimodal entity recognition. We adopt the entity F1-score as the evaluation metric. Fig. 7 demonstrates time, location and person extraction from chat logs.

Figure 7: An example of MM-NER dataset.

### A.2.4 MOBILE FUNCTION CALL

The Mobile Function Call (Mobile-FC) task is designed to evaluate the ability of MLLMs to call mobile API functions. The task requires the model to select appropriate functions from a given set of application functions to call according to the user's app instruction questions and output the parameters required for the function calls. We define 50 function call interfaces for different scenarios, such as setting an alarm, checking the weather, and setting navigation. The questions in the data are manually constructed by annotators, simulating real-world scenarios of apps on smartphone operating systems and forming complete multi-round dialogues. The evaluation method mainly compares the predicted function names and parameter names with the annotated results. A perfect match scores 1 point; otherwise, 0 points. As shown in the Fig. 8, we define the function name create alarm for setting an alarm, with the time field as the input parameter.

Figure 8: An example of Mobile-FC dataset.

### A.2.5 AGENT CONTEXT UNDERSTANDING

The Agent Context Understanding (ACU) task is used to assess the context-aware dialogue comprehension ability of MLLMs. The data is presented in the form of multi-turn conversations (including text and image). The model is required to resolve the anaphoric information in the user's final question based on multi-turn conversations or image information, and output a question that contains no anaphora. As shown in the Fig. 9, the user asks "Do you like his songs?". If no image is provided, the model needs to determine who "he" refers to based on the historical conversation. Otherwise, the model needs to recognize the person in the image. Model's output is a question that contains no referential information. We use the BLEU of the output answer with the reference answer to evaluate task performance, with scores ranging from 0 to 1.

**User**: Do you know Jay Chou?

**Assistant**: I know Jay Chou. He is a highly influential male pop music singer.

**User**：Do you like his songs?

**Answer**: Do you like Jay Chou's songs?

(a) Pure-text conversation sample.

**User**: Do you like his songs?

**Answer**: Do you like Jay Chou's songs?

(b) Multimodal conversation sample.

Figure 9: Examples of ACU dataset.

Table 8: Training dataset sizes.

| Dataset | Source | Data size | Dataset | Source | Data size |
|---------|--------|-----------|---------|--------|-----------|
| MMIE | self-built | 1.8k | APP-Rec | self-built | 22.8k |
| MMU | self-built | 21.1k | RR | self-built | 10.5k |
| TP | self-built | 26.8k | FC | self-built | 10.4k |
| ITR | self-built | 9.7k | ShareGPT4 | open-source | 36k |
| NER | open-source | 8k | Infinity-MM | open-source | 37.2k |
| OCR | open-source | 33k | SuperCLUE-Agent | open-source | 1.5k |

### A.2.6 APP RECOGNITION

The APP Recognition (APP-Rec) task, similar to the APP-Rec training set, is used to evaluate the ability of MLLMs to identify mobile applications. The model is required to directly output the APP name based on the content of the input mobile APP interface image, as illustrated in the Fig. 11. The performance evaluation is conducted by comparing the overlap between the predicted application name and the annotated result. A correct prediction scores 1 point; otherwise, 0 points.

### A.3 TRAINING DATASETS

The open source data includes: ShareGPT4 shibing624 (2023), NER (composed of Chinese-NER-SFT qgyd2021 (2024a), Sentiment-Analysis Abhishek Shrivastava (2023), and Few-Shot-NER-SFT qgyd2021 (2024b)), Infinity-MM Gu et al. (2024), OCR (consisting of Vision-OCR-Financial-Reports-10K Hamed Rahimi (2024), Arxiv-OCR-v0.1-SFT Niccolò Zanichelli (2024) and Invoices-and-Receipts-OCR-v1 minyang (2024)), and SuperCLUE-Agent Liang Xu (2024).

The self-built datasets include MultiModal-Instruction-Evolution (MMIE), APP Recognition (APP-Rec), Reference-Resolution (RR), MultiModal-Understanding (MMU), Function-Calling (FC), Task-Planning (TP), and Image-Text-Relevance (ITR), which are primarily derived from data synthesis and real-world industrial scenarios. The size of samples in all the training data is shown in Table 8.

### A.3.1 MULTIMODAL INSTRUCTION EVOLUTION

The Multimodal Instruction Evolution (MMIE) task consists of 1.8K pieces of multimodal question-answering data. As shown in Fig. 10, given an initial query and image with several available tools, the methodology requires the model to generate more sophisticated and diversified questions. The generation pipeline comprises six structured phases:

- **Intent analysis**: Analyze the user's potential needs from multiple perspectives.

- **Scenario expansion**: Expand the scenario to increase the diversity and complexity of the initial question.

- **Task decomposition**: Decompose the scenario into multiple subtasks which can be executed correctly by provided tools.

- **Raise new question**: Propose a new question based on the expanded scenario and subtasks.

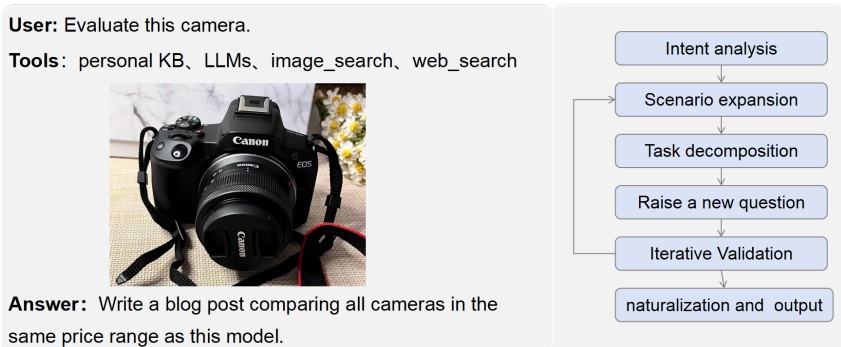

Figure 10: An example of MMIE dataset.

- **Iterative Validation**: Evaluate completeness and complexity, where completeness indicates whether the question adequately covers the steps of the subtasks.

- **Naturalization and output**: Refine questions to be more colloquial and output the final result.

### A.3.2 APP RECOGNITION

The APP Recognition (APP-Rec) task consists of 22.8K pieces of multimodal question-answering data, which are composed of images and task instructions. The task requires the model to identify the interface information of mobile apps in the input images and directly output the app names. To obtain diverse app interface data, we install 100 different applications on a mobile phone, such as WeChat, QQ, Little Red Book, Weibo, Alipay, Pinduoduo, Taobao, and TikTok. Annotators are then required to manually capture screenshots of different functional interfaces of each application, which serve as the image source for the APP-Rec task, as illustrated in Fig. 11. The default input task instruction is "Identify which app the screenshot belongs to?", and the answer is the name of the app corresponding to the image.

### A.3.3 REFERENCE RESOLUTION

The Reference Resolution (RR) task corresponds to the MM-RR task in Section A.2.2 which contains 10.5K pieces of multimodal question-answering data. We collect various images containing text information from the internet, with sources including academic papers, test questions, news, company official websites, Wikipedia, etc. Annotators design corresponding questions based on the text content in the given images as positive examples, while negative examples are obtained by replacing the images with different types, as shown in the following Fig. 6, which provides one positive and one negative example respectively.

### A.3.4 MULTIMODAL UNDERSTANDING

The Multimodal Understanding (MMU) tasks are consistent with ACU tasks in Section A.2.5. It takes the form of multimodal or text-only multi-round dialogues, with 1-4 rounds and a total of 21.1K samples. The images are sourced from publicly available internet data, covering various fields such as people, animals, plants, architecture, and digital products. The dialogue data is manually constructed by annotators based on the given images, focusing on reference problems. The task requires the model to combine the images and historical dialogue content to rewrite the user's final input text. This is achieved by replacing pronouns or supplementing omitted content to make the text semantically complete.

### A.3.5 FUNCTION CALLING

The Function Calling (FC) task consists of plain text instructions, which requires selecting appropriate tools from given tool set and filling in correct parameters for executing. The tools involve practical mobile applications such as unit conversion, weather inquiry, time calculation, text creation, recipe

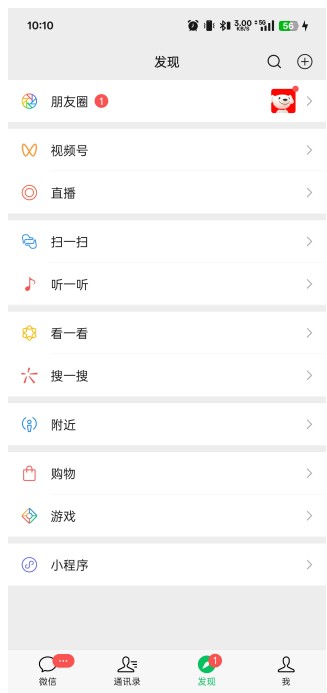

Figure 11: An example of APP-Rec dataset.

search, mobile phone bill inquiry, and other 500 types of useful tools. Notably, 90% of the instructions only require the invocation of a single tool.

Here is an example in Fig. 12: The user inquires how much 500 US dollars is in Japanese yen, and the answer includes thoughts and actions. The thought process briefly outlines the current step, while the action first provides the name of the selected tool and sets the actual parameters in the action input.

**User**: I am planning a trip to Japan next month and wonder how much Japanese yen I can exchange for 500 USD.

```
Answer: [
        {
            "Thought": "Check how many Japanese yen can be exchanged for 500 USD.",
            "Action": "exchange_rate",
            "Action Input": {
                "money": "500",
                "fromcoin": "USD",
                "tocoin": "JPY"
            }
        }
    ]
```

Figure 12: An output example of FC dataset.

### A.3.6 TASK PLANNING

The Task Planning (TP) dataset, also targeting tool calling scenarios, places greater emphasis on multi-stage operations with inter-dependent steps compared to FC. It involves 26.8K pieces of multimodal question-answering data. This dataset requires models to properly decompose complex problems into solvable subtasks while ensuring correct tool selection and execution. In multistep scenarios, managing inter-parameter dependencies becomes critical.

The input is a complex question requiring calling apps on moblie phone. Output contains multistep thinking and actions similar to FC, and symbols start with #E are used to receive parameter for cited in subsequent tasks. (as demonstrated in Fig. 2).

### A.3.7 IMAGE-TEXT RELEVANCE

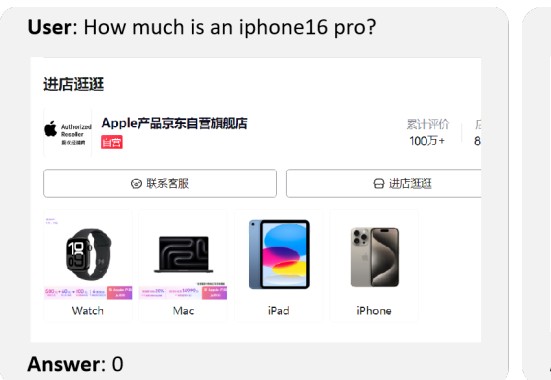
(a) A negative sample example.

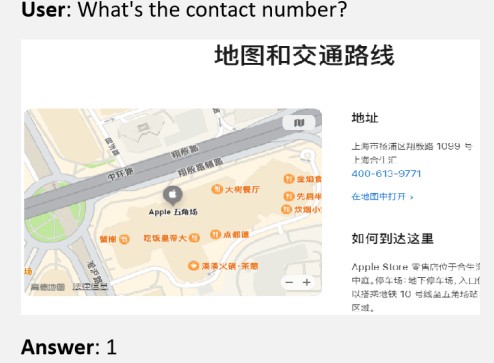
(b) A positive sample example.

Figure 13: Examples of ITR dataset.

The Image-Text Relevance (ITR) task involves 9.7K pieces of multimodal question-answering data. The task requires the model to analyze the relevance between the question and the image based on the characteristics of the question. If the image is relevant to the question and can be used to answer the user's question, the model should answer 1; otherwise, 0. The images are sourced from publicly available internet data, the same as those used in the MMU task. Annotators manually construct questions related to the image content as positive examples. For example, for images of people, questions about names, works, or family relationships can be asked. Negative examples are constructed by replacing the images with different types, as shown in the following Fig. 13, which presents one positive and one negative example respectively.

## B ALGORITHM

---

**Algorithm 1:** Algorithm of DaMo

---

**Input:** $\mathcal{D}$: training dataset; $\mathcal{D}^{test}$: test dataset; $\theta_0$: initial parameters of MLLM; $\mathcal{P}$: data mixing space, consisting of data mixture $\mathbf{p}$; $f_{MLP}$: fitted MLP; $t$: training steps; $\mathcal{M}$: The data points for fitting MLP, consisting of pairs <($\mathbf{p}$, t), $\mathbf{s}$>.

**Output:** $\theta^*$: MLLM trained with the optimal data mixture $\mathbf{p}^*$.

Initialize $\mathcal{M} \leftarrow \emptyset$

Randomly sample a small subset $\mathcal{P}_{mlp} \subset \mathcal{P}_{fix}$

**foreach** $\mathbf{p}^i \in \mathcal{P}_{train}$ **do**

    $\theta_t^i \leftarrow \text{Trainer}(\mathcal{D}, \mathbf{p}^i, t, \theta_0)$

    $\mathbf{s}^i \leftarrow \text{Evaluator}(\theta_t^i, \mathcal{D}^{test})$

    $\mathcal{M} \leftarrow \mathcal{M} \cup \{(\mathbf{p}^i, t, \mathbf{s}^i)\}$

**end**

$f_{MLP} \leftarrow fit(\mathcal{M})$

$\mathbf{p}^*, t^* \leftarrow \arg\max_{\mathbf{p} \in \mathcal{P}_{fix}} f_{MLP}(\mathbf{p}, t)$

$\theta^* \leftarrow \text{Trainer}(\mathcal{D}, \mathbf{p}^*, t^*, \theta_0)$

$\hat{\mathbf{s}} \leftarrow \text{Evaluator}(\theta^*, \mathcal{D}^{test})$

**return** $\theta^*$, $\mathbf{s}^*$

---

## C   EXTENSIBLE VALIDATION

We validate the generalizability of DaMo across model families and model sizes (Qwen2.5VL-3B, Qwen2.5VL-7B, InternVL3-14B) through a two-stage sampling strategy: 1) Random selection from base model's experimental mixtures, and 2) Strategic sampling from extrapolated optimal mixtures. We plot the predicted overall average scores of target models(using the original DaMo) on the x-axis against the ground-truth overall average scores on the y-axis. If mixture $\mathbf{p}_i$ always outperforms mixture $\mathbf{p}_j$ for any $i, j$ across models (Pearson correlation coefficient $r = 1$), it proves that DaMo has perfect transferability. As shown in the upper panel of Fig. 5, the Pearson correlation coefficients are consistently above 0.75, demonstrating robust cross-model applicability of DaMo. This suggests that optimal mixtures identified for base model likely remain near-optimal for target models.

Although DaMo demonstrates promising cross-model transferability, model variations still introduce errors when extrapolating the optimal mixture for target models. To address this, we establish a linear mapping $g = f()W + b$ between base model's DaMo $F$ and target model's optimal law $G$ using 20 calibration samples. This projection improves Pearson correlation to 0.90(bottom panel of Fig. 5), enabling more precise extrapolation. As Table 5 demonstrates, the linear-mapped DaMo achieve superior performance compared to direct extrapolation from the base model's DaMo.

## D   LIMITATIONS AND FUTURE WORK

Our study is grounded in two key assumptions: (1) disregarding sample order within individual datasets, and (2) maintaining a fixed data mixture ratio throughout training. While recent research reports the efficacy of multi-stage training and curriculum learning, our preliminary attempts to relax these assumptions—specifically through dynamic data mixture adjustments—remain exploratory. We have yet to establish a systematic methodology for extrapolating optimal dynamic mixtures or quantify the computational costs and performance gains relative to fixed data mixture.

Moving forward, we plan to formalize a framework for dynamic data mixture optimization. This will involve integrating Monte Carlo Tree Search (MCTS) with reinforcement learning to iteratively determine stage-specific data mixtures, aiming to extrapolate optimal mixture trajectories that mitigate task conflict and catastrophic forgetting. Additionally, we propose incorporating sample quality metrics as input variables for downstream performance prediction, enabling difficulty-aware sampling during training. Further, we will enhance PhoneAgentBench to better align with the fast-evolving requirements of on-device AI deployment, ensuring its adaptability to emerging mobile-centric AI paradigms.

## E   LLM USAGE

In this paper, we used LLMs to polish the content of the main text and appendices.

