# OpenReview forum: "DaMo: Data Mixing Optimizer in Fine-tuning  Multimodal LLMs for Mobile Phone  Agents"
_ICLR.cc/2026/Conference — ICLR 2026 Conference Withdrawn Submission_

### Official Review · Reviewer_rXkT · 2025-10-29

**Soundness:** 2
**Presentation:** 3
**Contribution:** 2
**Rating:** 6
**Confidence:** 3

**Summary:**

The paper introduces DaMo (Data Mixture Optimizer), a learned MLP that predicts downstream task performance for a given data-mixture ratio and training step, enabling selection of near-optimal mixtures for multitask SFT of multimodal LLMs without exhaustively fine-tuning the base model. The authors also propose PhoneAgentBench, a multimodal benchmark for mobile phone agents that covers complex task planning, device-native tool use, multimodal memory, and screen-context understanding. Empirically, DaMo outperforms heuristic mixtures (uniform/natural) and a loss-fitting baseline (DML) on PhoneAgentBench and several general benchmarks (e.g., BFCL-v3, MME-Perception/Reasoning, OCRBench); the predictor attains strong fit and transfers across model families with light calibration.

**Strengths:**

1. The problem of looking for optimal data mixture of SFT with minimal effort is practical.
2. The proposed benchmark enhances the complexity of the MLLM phone agent task making the evalutation more approaching real world settings
3. The result shows that applying this method can effectively improve the performance of finetuning for different benchmarks

**Weaknesses:**

1. There is disconnection between the proposed data mixing strategy and benchmark. The DaMo is for the multi-task SFT in terms of MLLM in phone agent tasks. The benchmark is more on the phone agent side with mixed problem.
2. Although the claim lies on the no training of LLM in the model, the data collection process should be done repeatedly to collect mixture rate-performance pair for training the MLP optimizer. It involves querying MLLM for the tasks. The cost of this process is neglectable and should be audited to give better understanding of this approach. From paper, it requires 250 mixtures x 4 checkpoints to fit predictor at 250 samples. Table 2 reports 4225 H20-hours even in 250 sample setting. This cost may be prohibitive for many labs and partially offsets the claimed efficiency.
3.  Static mixture shows the limitation of the model updating. As to find optimal mixture in a global view, the MLLM inference is based on the initial state and MLP fixed once mixture is explored. The training involves parameter change with MLLM.
4.the scalability and coverage constraint is not clear. It is not unclear how DaMo scales when 𝑚 and domain diversity grow further, or when noisy/long-tail domains dominate.

**Questions:**

1. Given the reported ~4,225 H20-hours to fit the predictor at 250 samples, how does total wall-clock and dollar cost compare to a strong black-box search (e.g., BO with early stopping) under the same budget? Any ablation on fewer mixtures or adaptive sampling strategies?
2. How sensitive is MLP fitted to the initial state $\theta_{0}$.If we change pretraining mixtures or use a different instruction-tuned seed, do we need to refit from scratch? Could the 20-sample linear calibration be replaced with a zero-shot normalization?
3. For MT-Plan, does the “longest ordered action sequence” correlate with execution success on device? Any human eval or simulated execution to validate that higher scores lead to fewer task-failures/corrections?
4.  In the spaces of P, if m grows the space can explode. Do you foresee the current sampling regime remaining effective, or is a sparse/structured parameterization of p (grouped datasets, low-rank factors) needed?
5. You show transfer with a 20-sample linear map; what happens for substantially different domains (e.g., more OCR-heavy or audio-centric phone tasks)? How many calibration samples are needed before diminishing returns?

---

### Official Review · Reviewer_ajXq · 2025-10-29

**Soundness:** 2
**Presentation:** 2
**Contribution:** 2
**Rating:** 2
**Confidence:** 3

**Summary:**

This paper proposes DaMo (Data Mixing Optimizer), a method for optimizing data mixtures during supervised fine-tuning (SFT) of multimodal large language models (MLLMs) for mobile phone agent tasks. The approach uses a trainable multi-layer perceptron (MLP) to predict downstream task performance given any data mixture ratio, enabling efficient identification of optimal data combinations without exhaustive training. The authors introduce PhoneAgentBench, a new benchmark with 1,235 QA pairs covering six tasks: multimodal task planning, app recognition, reference resolution, context understanding, named entity recognition, and function calling. DaMo achieves 3.38% improvement on PhoneAgentBench over baseline methods and demonstrates transferability across different model architectures (Qwen2.5VL-3B/7B, InternVL3-14B)

**Strengths:**

1. Practical problem formulation: Addresses a real industrial need for optimizing multi-dataset fine-tuning, which is highly relevant for practitioners working with limited computational budgets.

2. Novel use of neural networks for data mixing: The shift from exponential/power-law functions (used in pretraining literature) to trainable MLPs for SFT performance prediction is well-motivated through empirical analysis (Figure 3)

**Weaknesses:**

1. Extremely limited algorithmic novelty: The core contribution is fitting an MLP to predict performance—essentially a standard regression task. The MLP architecture is trivial (2 hidden layers, 100 neurons each from sklearn). This is purely an engineering/empirical contribution with no theoretical insight into why this works or when it might fail.

2. Weak theoretical justification: Why should an MLP with 100 hidden units be sufficient to model the performance landscape of a 4B parameter model across 12 datasets?
> No analysis of what makes certain data mixtures transferable across models.
> The assumption of fixed mixtures throughout training is very restrictive and may be suboptimal (acknowledged in Appendix D but not addressed).

3. Experimental limitations: Only evaluated on one base model architecture (InternVL2.5-4B) for fitting; transferability experiments use limited calibration (20 samples).
> No ablation on MLP architecture choices—would a simpler linear model suffice? Would a deeper network improve $R^2$?
> The $R^2$=0.81 fitting score means ~19% unexplained variance, how would you judge this number? As far as I understand, this could still lead to suboptimal mixture selection.

**Questions:**

- MLP architecture justification: Why is a 2-layer MLP with 100 hidden units the right choice? Have you tried simpler models (linear regression, polynomial regression) or more complex architectures (deeper MLPs, transformers)? What is the sensitivity analysis?

- Sample efficiency: You claim 250 samples are sufficient, but Table 2 shows $R^2$ improving from 0.58 (50 samples) to 0.81 (250 samples). Have you tried 500 or 1000 samples? Where is the diminishing returns threshold?

- Generalization beyond domain: You evaluate on PhoneAgentBench and general benchmarks. How would DaMo perform on:
    1. Completely different downstream tasks not seen during MLP fitting?
    2. Non-agent applications (e.g., visual question answering, image captioning)?

---

### Official Review · Reviewer_e18z · 2025-10-30

**Soundness:** 3
**Presentation:** 3
**Contribution:** 3
**Rating:** 4
**Confidence:** 4

**Summary:**

This paper studies the problem of choosing optimal training-data mixture ratios for supervised fine‑tuning (SFT) of multimodal large language models (MLLMs) targeted at mobile phone agent tasks. The authors propose DaMo (Data Mixing Optimizer), which learns a predictive model f(p, t) that maps a fixed mixture proportion p (over m datasets) and training step t to downstream task performance s. f is implemented as a small MLP trained from a relatively small set of pilot runs on a base MLLM (InternVL2.5‑4B). After fitting, DaMo exhaustively predicts performance over the discrete fixed‑mixture space . The paper also introduces PhoneAgentBench, a suite of phone‑agent tasks (planning, tool use, memory, screen understanding).

**Strengths:**

1. Addressing data‑mixture selection for SFT of MLLMs is an underexplored but important problem for multi‑task agents. The proposed learn‑and‑extrapolate pipeline is conceptually simple and practical
2. The paper demonstrates nontrivial predictive performance (R²≈0.81 with 250 samples), shows end‑to‑end gains across a new PhoneAgentBench and several public benchmarks
3. PhoneAgentBench (multi‑dimensional tasks: planning, tool invocation, memory, screen understanding) fills a gap versus prior GUI‑focused or single‑dimension agent benchmarks and is a useful contribution if released with sufficient documentation.

**Weaknesses:**

1. There are contradictory statements about PhoneAgentBench size. It is unclear what exact splits were used for fitting the predictor versus final evaluation.
2. Obtaining the 250 fit samples reportedly costs thousands of H20‑GPU hours (Table 2), which is nontrivial and may limit accessibility. The paper does not explore more sample‑efficient acquisition strategies.
3. It is not fully clear whether the downstream datasets used to fit f were held out from final evaluation (risk of overfitting the evaluation metrics). Baselines are limited to uniform/natural heuristics and an exponential loss fit (DML); comparison to other dataset‑weighting approaches (e.g., Group‑DRO, gradient‑based weighting, simple grid search with multiple seeds) is missing.

**Questions:**

1. For the key comparisons (Tables 3–5), please report (a) number of random seeds / repeated runs and standard deviations, (b) the exact training recipe used for baselines (was DML reimplemented/tuned for SFT?), and (c) a more precise compute accounting (total GPU‑hours for pilot sampling, for final training per mixture).
2. How were the 250 pilot mixtures sampled from Pf ix (purely uniform random, stratified, or other)? Did you try active sampling or Bayesian optimization to reduce the required number of pilot runs?

---

### Official Review · Reviewer_Mot7 · 2025-10-31

**Soundness:** 2
**Presentation:** 3
**Contribution:** 2
**Rating:** 4
**Confidence:** 2

**Summary:**

The paper introduces DaMo, a method for optimizing data mixture ratios in multitask supervised fine-tuning of multimodal LLMs for mobile phone agent tasks. The key innovation is using a trainable MLP to predict downstream task performance for arbitrary data mixtures, rather than relying on exponential/power-law functions used in pre training literature. The authors train the MLP on ~250 randomly sampled data mixtures, then extrapolate to find optimal mixtures from a large discrete space. The paper also introduces PhoneAgentBench, a benchmark with 1,235 test cases across 6 tasks evaluating mobile agent capabilities.

**Strengths:**

S1. Problem is clearly motivated, timely and practical, with clear demonstration that pre-training methods (exponential fitting) fail for multitask SFT.

S2: Strong performance on few benchmarks. For example, BFCL-v3 shows 18% absolute gain.

**Weaknesses:**

W1: My major concern is that I don't see any theoretical support for the generalization. The claim that 250 samples suffice for 17M mixtures, has no theoretical support.

W2. The whole process looks so expensive. 4,225 GPU-hours collecting 250 samples to train the MLP. But what if you just trained 250 random mixtures and picked the best one? Or trained 100 random mixtures and picked the best? The paper never compares against this obvious baseline, so we don't know if the MLP complexity adds value.

W3. Three tasks (MT-Plan, ACU, APP-Rec) have only 100 test samples. With improvements of 2-3%, these could be within noise. No confidence intervals, p-values, or multiple runs reported.

Minor:

---
W4. I don't see any inter-annotator agreement or human baseline.

**Questions:**

Q1. How does 4,225 GPU-hours for DaMo compare to: (a) random search with 100 samples, (b) Bayesian optimization with 50 evaluations, (c) grid search? What's the break-even point where DaMo becomes worthwhile?

Q2. With 100-sample test sets and 2-3% improvements, how significant are these results?

Q3. Why 2-layer MLP with 100 neurons specifically? Have authors tried other architectures?

---

### Note · Authors · 2026-01-06

I have read and agree with the venue's withdrawal policy on behalf of myself and my co-authors.